# How Does Cancer Occur? How Should It Be Treated? Treatment from the Perspective of Alkalization Therapy Based on Science-Based Medicine

**DOI:** 10.3390/biomedicines12102197

**Published:** 2024-09-26

**Authors:** Reo Hamaguchi, Masahide Isowa, Ryoko Narui, Hiromasa Morikawa, Toshihiro Okamoto, Hiromi Wada

**Affiliations:** 1Japanese Society on Inflammation and Metabolism in Cancer, 119 Nishioshikouji-cho, Nakagyo-ku, Kyoto 604-0842, Japan; rhamaguchi@scim.or.jp (R.H.); misowa@scim.or.jp (M.I.); rnarui@scim.or.jp (R.N.); hmorikawa@scim.or.jp (H.M.); 2Department of Thoracic and Cardiovascular Surgery, Cleveland Clinic, Cleveland, OH 44195, USA; okamott@ccf.org

**Keywords:** cancer metabolism, mitochondrial dysfunction, Warburg effect, alkalization therapy, tumor microenvironment, pH regulation, glycolysis, oxidative phosphorylation

## Abstract

This review article investigates the relationship between mitochondrial dysfunction and cancer progression, emphasizing the metabolic shifts that promote tumor growth. Mitochondria are crucial for cellular energy production, but they also play a significant role in cancer progression by promoting glycolysis even under oxygen-rich conditions, a phenomenon known as the Warburg effect. This metabolic reprogramming enables cancer cells to maintain an alkaline internal pH and an acidic external environment, which are critical for their proliferation and survival in hypoxic conditions. The article also explores the acidic tumor microenvironment (TME), a consequence of intensive glycolytic activity and proton production by cancer cells. This acidic milieu enhances the invasiveness and metastatic potential of cancer cells and contributes to increased resistance to chemotherapy. Alkalization therapy, which involves neutralizing this acidity through dietary modifications and the administration of alkalizing agents such as sodium bicarbonate, is highlighted as an effective strategy to counteract these adverse conditions and impede cancer progression. Integrating insights from science-based medicine, the review evaluates the effectiveness of alkalization therapy across various cancer types through clinical assessments. Science-based medicine, which utilizes inductive reasoning from observed clinical outcomes, lends support to the hypothesis of metabolic reprogramming in cancer treatment. By addressing both metabolic and environmental disruptions, this review suggests that considering cancer as primarily a metabolic disorder could lead to more targeted and effective treatment strategies, potentially improving outcomes for patients with advanced-stage cancers.

## 1. Introduction

Eukaryotic multicellular organisms, including humans, possess mitochondria within their cells. Mitochondria typically generate energy through oxidative phosphorylation (OXPHOS) within the cell [1]. Cancer is a disease characterized by the abnormal proliferation of cells within our bodies, triggered by various factors. Recent studies have begun to illuminate the relationship between mitochondria and the onset of cancer. From a biosynthetic perspective, mitochondria are intracellular organelles whose primary function is energy production, serving as the cell’s power plants. They are also known to play a crucial role as a type of suicide weapon. Indeed, mitochondria have several dozen lethal signal transduction pathways, particularly when the permeability of the mitochondrial outer membrane increases and pro-apoptotic proteins are released into the cytoplasm, causing dysfunction in mitochondrial bioenergetic functions. This leads to the promotion of cell self-destruction, known as apoptosis. Cancer cells are reported to exhibit metabolic abnormalities in mitochondria. Normally, mitochondria produce energy through OXPHOS within the cell; however, in cancer cells, glycolytic intermediates are used in anabolic reactions as substrates for cell proliferation, thus impairing their precise control [2]. Furthermore, cancer tends to occur in organs with hypoxic conditions. This is because mitochondria use oxygen for energy production, and in hypoxic conditions, mitochondria are destroyed, leading to apoptosis of the cell. However, cancer cells evade this apoptosis and choose to survive instead. Thus, it is suggested that abnormalities in energy production in mitochondria and the evasion of apoptosis are deeply involved in the onset of cancer.

Mitochondrial DNA contains information mainly related to mitochondrial proteins and urgently required information at the site where mitochondria are replicated during mitosis. Much of the information necessary for mitochondrial production is contained in nuclear DNA, and mitochondria cannot exist outside the cell. Rather, mitochondria play a role in producing the energy needed by cells using oxygen, and eukaryotic cells themselves cannot survive without mitochondria. These facts support the hypothesis that mitochondria originate from intracellular symbiosis. Approximately two billion years ago, eukaryotes that emerged from archaebacteria engulfed proteobacteria. Within the host cell, proteobacteria transformed into mitochondria, leading to the evolution of fungi and animals. Conversely, the cells that engulfed proteobacteria took in cyanobacteria about one billion years ago, and cyanobacteria transformed into chloroplasts, leading to the evolution of algae and plants. Intracellular symbiosis refers to the phenomenon where a host organism incorporates a different species of prokaryote, forming intracellular organelles with chloroplasts and mitochondria [3].

Entropy, as a fundamental concept in thermodynamics, represents disorder and randomness, offering valuable insights into tumor behavior and therapy. Recent studies have suggested that entropy not only characterizes the disordered state of cancer cells but also correlates with various aspects of tumor pathology, including progression and response to treatment. For instance, network entropy, which measures the disorder within a cell’s molecular interaction network, is increased in cancer cells. This increased network entropy suggests more chaotic molecular interactions, which are associated with higher metastatic potential and therapeutic resistance [4,5,6]. Similarly, the parallel increase in genomic and transcriptomic entropy as cancer advances points to the systemic nature of this disorder, affecting both DNA structure and gene expression [5,6]. Moreover, entropy measurements have shown to be useful in assessing tumor heterogeneity and prognosis, where higher entropy levels correlate with increased tumor complexity and poorer patient outcomes [6,7,8,9]. The incorporation of entropy into cancer research provides a new perspective on the complexity of cancer and suggests potential predictive value for tumor behavior and response to therapy, thereby enriching our understanding of cancer as a thermodynamically active system [7,8,10,11].

This review summarizes the roles of mitochondrial dysfunction and the cancer-specific metabolic adaptations in enabling cancer cell survival in low-oxygen environments by maintaining distinct internal and external pH levels. The discussion also highlights how the acidic tumor microenvironment (TME) promotes cancer invasiveness and resistance to treatment. Furthermore, the review explores managing systemic entropy through alkalization therapy to neutralize the acidic TME and enhance cancer treatment outcomes using science-based medicine principles that employ inductive reasoning from observed clinical outcomes to support a metabolic re-evaluation of cancer therapy.

## 2. Why Does Cancer Arise in the Human Body?

Otto Warburg, a pioneer in cancer metabolism research, reported in a 1956 Science article titled “On the Origin of Cancer Cells” that cancer cells have the property of deriving energy from glycolysis (aerobic glycolysis) even in the presence of ample oxygen [12,13]. This phenomenon is commonly known as the Warburg effect. Warburg referenced an intriguing study reported in 1953 by Goldblatt and Cameron in the Journal of Experimental Medicine [14]. According to their study, intermittent anaerobic culturing of fibroblasts derived from the heart over extended periods resulted in enhanced cell proliferation and led to carcinogenesis. Cells that did not experience oxygen deprivation did not become cancerous. This suggests that the carcinogenesis of cells occurs under conditions of nutrient and oxygen deficiency.

The processes thought to occur within the cell at this time are as follows:(i)Even in the presence of local nutrients, the onset of hypoxic or anoxic conditions leads to cell death. This is believed to be due to the cessation of OXPHOS by mitochondria, resulting in mitochondrial dysfunction and a halt in ATP production.(ii)Subsequently, the energy charge needed to maintain mitochondrial membrane potential becomes unavailable, leading to mitochondrial collapse. This results in the release of cytochrome c and cell death [15].(iii)In the aftermath, cells that have adapted to a metabolism capable of functioning without oxygen emerge. It is believed that the signaling from mitochondria to the nucleus, activating metabolic switches, represents a mechanism known as the “retrograde response” [16,17,18,19].(iv)Such “switched on” cells use glycolysis (substrate-level phosphorylation) exclusively for energy metabolism and live “selfishly” without oxygen [20].(v)These cells are actively engaged in cellular activities to replicate for the next generation. At this time, the pH of cancer cells is always “alkaline” (likely, the Golgi apparatus and intracellular vesicles are strongly acidic) [21].

In other words, cancer cells metabolize using glycolysis even under aerobic conditions, forcing them to live selfishly [22,23,24,25].

## 3. What Characteristics Do Such Cancers Possess?

The excess protons produced by glycolysis are expelled out of the cell by pumps such as sodium-hydrogen exchanger isoform 1 (NHE-1), consequently creating a strongly acidic microenvironment around the cancer cells. In such conditions, the tumor acquires various characteristics as described below. In this acidic setting, a large number of bone marrow-derived suppressor cells are recruited to the Tumor Microenvironment (TME), induced by the protons present. This leads to a substantial increase in inflammatory cells, neutrophils, and macrophages [26,27,28,29,30,31].

The acidic TME promote the following traits:(i)Increased malignancy(ii)Enhanced activation of cell proliferation and division cycles(iii)Expression and activation of genetic abnormalities and oncogenes(iv)Activation of cell proliferation factors(v)Increased glycolytic activity(vi)Activation of DNA synthesis(vii)Activation of cell migration(viii)Stimulation of angiogenesis(ix)Increased metastatic potential(x)Activation of multidrug resistance genes

Within this acidic TME, tumors develop aggressive traits that enhance malignancy and complicate treatment. The acidic conditions promote rapid cancer growth through enhanced cell proliferation, increased glycolytic activity, and DNA synthesis. Additionally, they facilitate tumor spread by promoting angiogenesis and cell migration, while also stimulating multidrug resistance genes, thereby reducing the effectiveness of chemotherapy [32,33,34,35,36,37,38].

Moreover, the relationship between the TME and tumor immunity is closely intertwined [39,40,41]. It is reported that the acidic nature of the TME diminishes anti-cancer immune responses [42]. The acidic TME suppresses various immune cells such as dendritic cells, natural killer cells, cytotoxic T cells, and macrophages. This leads to reduced antitumor immune activities and facilitates cancer immune escape [43,44]. Furthermore, an in vitro study using mouse models of melanoma has shown that this acidic environment suppresses T-cell responses and decreases the secretion of immune signaling molecules IFN-γ and TNF-α [34]. The acidic TME increases tumor aggressiveness by enabling the cancer cells to evade immune surveillance and enhance their survival capabilities. Therefore, targeted therapies that can neutralize the acidic conditions may improve cancer treatment outcomes.

## 4. Dissipative Structures and Entropy

To reduce cancer cell aggressiveness and enhance the effectiveness of various therapeutic methods, detoxification of the cancer patient’s body is necessary. The fundamental concept underlying this requirement is that of “dissipative structures”. Certain interventions in dissipative structures can lead to a reduction in the system’s entropy. Within these nonequilibrium open systems, the interaction of matter and energy can lead to the organization of phenomena, seemingly contradicting the Second Law of Thermodynamics (the law of increasing entropy). Multicellular organisms, including humans, are examples of “dissipative structures in nonequilibrium open systems”, and cancer activity can be suppressed through specific interventions [45].

The essence of the Second Law of Thermodynamics, concerning the generation of entropy, rests on the critical distinction between “reversible” processes, which are theoretically capable of returning a system to its initial state without net energy dissipation, and “irreversible” processes, which increase the total entropy of a system, making a return to the initial state impossible without external intervention. This fundamental differentiation underpins the introduction of entropy *S* and the formulation of the Second Law. The Second Law asserts the existence of a function *S*, which increases monotonically until reaching its maximum at thermodynamic equilibrium. Applying this principle to systems that exchange energy and matter with their environment reveals that the change in entropy (dS) comprises two parts: the entropy transferred across the system’s boundaries (deS) and the entropy generated within the system (diS) (Figure 1). While the Second Law assumes that the internal generation of *S* is positive (or zero), indicative of the natural progression towards disorder, in the realm of dissipative structures—systems far from equilibrium that maintain a steady state through the continuous input and dissipation of energy—phenomena resembling negative entropy (entropy reduction) can manifest, challenging and enriching our understanding of thermodynamic principles [46].

As a result, these are steady structures that emerge as sources of self-organization within the energy flow, as proposed by Nobel laureate Ilya Prigogine. These structures, known as steady-state open systems or nonequilibrium open systems, differ fundamentally from rocks, which inherently maintain their stable structure. Dissipative structures, akin to whirlpools generated by the kinetic energy of tides in an inland sea, require specific inputs to sustain their structure. Since dissipative systems are open systems, entropy is maintained within a certain range, allowing for the exchange of energy both internally and externally. It is particularly intriguing that biological phenomena, such as cellular metabolic processes and ecological systems, can be understood as open systems in a steady state, highlighting the applicability of nonequilibrium thermodynamics beyond traditional physics. While equilibrium thermodynamics once dominated the field, the focus on steady states in nonequilibrium thermodynamics has opened new avenues for understanding complex systems, from meteorological patterns to the intricate behaviors of living organisms. This shift towards nonequilibrium thermodynamics underscores the evolving nature of scientific inquiry, emphasizing the dynamic interactions that characterize the natural world [45].

The interpretation of biological phenomena as dissipative structures offers significant insights. In this framework, humans consume and digest food, with carbohydrates serving as the primary energy source. Following glycolysis, oxygen is utilized in OXPHOS to extract the bulk of energy [1]. The residues of digestion are then expelled as waste, exemplifying the system’s entropy exchange (deS) and the generation of entropy within (diS). Intriguingly, every living organism possesses a “lifespan”, during which entropy accumulation within this dissipative structure may lead to its inactivity and eventual demise. Optimal nutrition and self-care practices have been shown to potentially extend the human lifespan to around 120 years, with premature death often attributed to excessive internal entropy accumulation [47]. This framework similarly applies to cancer treatment. For example, a body burdened with excess metabolic byproducts is more prone to cancer development. Removing this entropy can, therefore, potentiate treatment efficacy and increase recovery chances. In the following section, we delve into alkalization therapy, a novel treatment method we propose based on this framework [48,49]. Alkalization therapy, grounded in the principles of dissipative structures, aims to reduce the body’s entropy load, thereby fostering a more conducive environment for recovery and health restoration. This approach aligns with emerging scientific perspectives that advocate for a holistic understanding of disease mechanisms, particularly emphasizing the critical role of systemic entropy in the progression of health and disease.

## 5. The Relationship between Cancer and pH, and Alkalization Therapy

While the intracellular pH (pHi) of normal cells is neutral (ranging from 6.9 to 7.1), the pHi of cancer cells tends to be alkaline (between 7.2 and 7.7) [50]. In normal cells, NHE-1 remains inactive, serving as a set point for proton-dependent activation, as the pHi typically resides within the physiological resting range of 6.9 to 7.1. This resting pHi is primarily maintained by Na^+^-driven Cl^−^/HCO_3_^−^ exchangers, which play a role in sustaining the pHi above the predicted equilibrium. Conversely, Na^+^-independent Cl^−^/HCO_3_^−^ exchangers regulate the intracellular Cl^−^ concentration, restoring pHi to its physiological resting value. NHE-1 and Na^+^-independent Cl^−^/HCO_3_^−^ exchangers act as “housekeepers” of pHi, being activated only when the cell becomes acidified or alkalized, respectively, thus regulating the homeostasis of pHi. This regulation relies on the cell’s buffering capacity and various membrane ion exchangers, including the NHE-1, both Na^+^-dependent and Na^+^-independent Cl^−^/HCO_3_^−^ exchangers, and monocarboxylate transporters [51].

Due to their unique metabolic traits, cancer cells frequently induce hypoxia, particularly in poorly perfused regions and extracellular acidosis. These cells have adapted to thrive in more acidic environments compared to normal cells. A crucial mechanism in this evolutionary adaptation involves the activation of proton transporters, such as NHE-1. These transporters are instrumental in regulating intracellular pH by expelling excess protons produced during glycolysis, thereby lowering the extracellular pH and acidifying the cancer cells’ surrounding environment [49,52,53,54,55]. Recent research has shed light on the profound influence of signaling pathways, altered by genetic mutations and the tumor microenvironment, on the central metabolism. In tumors, the intracellular metabolism primarily initiates rapid energy production through aerobic glycolysis, leading to swift ATP generation. This metabolic activity is coupled with the synthesis of “macromolecules”, crucial for the development of subsequent cell generations. These biochemical reactions are meticulously regulated by the redox state, thus amplifying metabolic activity and glycolysis and culminating in the significant production of protons (H^+^) within the cell. These protons are then expelled from the cell via activated proton transporters, contributing to the acidic extracellular environment characteristic of cancerous tissues [21,56,57,58,59,60,61,62,63,64,65,66].

Therapeutic approaches that target this pH regulator constitute our proposed Alkalization therapy [49]. Neutralizing the acidity of cancer cells may be associated with suppression of cancerous activity [34,67,68,69,70]. Alkaline pHi (acidic pHe) of cancer cells can diminish the intracellular uptake of many chemotherapeutic agents, including vinblastine, adriamycin, cisplatin, paclitaxel, and camptothecin, etc. A notable example is the reported increase in adriamycin resistance by 2000 times when the pHi rises from 7 to 7.4 in human lung cancer cells [67]. It is observed that most cancer patients treated with alkalization therapy in our clinic who showed improvement had an alkaline urinary pH. While currently, there is no clinical methodology available for directly measuring the pH of cancer cells, several imaging techniques such as ^31^P-magnetic resonance spectroscopy, ^89^Zr-labeled pH-low insertion peptide with positron emission tomography, and acido-chemical exchange saturation transfer magnetic resonance imaging are promising for assessing the pH of the TME [71,72,73,74]. Additionally, venous blood gas analysis for base excess (BE) serves as a more precise indicator, potentially reflecting systemic alterations influenced by the tumor’s metabolic activities. Although urinary pH might have some utility for monitoring pH changes, it is essential to assess carefully whether it accurately reflects the pH of the TME, as the association remains uncertain. Continued alkalinization of urine may reflect systemic alkalinization, but this should be interpreted with caution. Alkalization of the acidic TME may reduce the activity of NHE-1, which plays a critical role in maintaining an alkaline intracellular environment that supports cancer cell proliferation. Therefore, targeting NHE-1 activity through alkalization could enhance the effectiveness of treatments and ultimately lead to improved patient outcomes [48,62,75,76,77,78,79,80].

Evidence-Based Medicine (EBM) primarily relies on deductive reasoning, starting from general hypotheses to predict specific outcomes, which can limit the exploration of new knowledge. This is particularly evident in the treatment of stage 4 cancer, where EBM often assumes incurability. Science-Based Medicine (SBM), on the other hand, uses inductive reasoning to learn from unique cases, such as individuals recovering from stage 4 cancer, to generate new hypotheses for potential treatments. Alkalization therapy, for instance, has been developed through an inductive method based on scientific facts demonstrated in basic medicine—such as the relationship between cancer and pH, and cancer metabolism—as well as data from patients who have survived stage 4 cancer. By integrating both approaches, we can challenge existing assumptions within EBM and expand the possibilities for curing complex diseases like stage 4 cancer.

## 6. Clinical Studies of Alkalization Therapy in Cancer Treatment

Alkalization therapy targets the acidic TME, a key feature of cancer metabolism that contributes to cancer progression and resistance to treatment. This therapy utilizes alkalizing agents like sodium bicarbonate and/or sodium potassium citrate, along with dietary interventions, and has demonstrated efficacy in suppressing cancer progression and enhancing the response to anti-cancer drugs in both in vitro and in vivo studies [48]. Recently, a phase I/II clinical study evaluated the safety and efficacy of an alkalizing agent combined with chemotherapy in metastatic pancreatic cancer patients, suggesting a potential survival benefit when the agent is used with S-1 as third/fourth-line therapy [81]. Although clinical research on alkalization therapy remains limited, we have reported case studies demonstrating favorable outcomes in patients with renal cancer, malignant lymphoma, gastric cancer, esophagogastric junction cancer, and breast cancer treated with this therapy [49,82]. Additionally, we have conducted retrospective studies investigating the impact of alkalization therapy on pancreatic cancer, small-cell lung cancer, and hepatocellular carcinoma. This section outlines these retrospective studies, highlighting the therapy’s effects across these varied conditions.


(i)Pancreatic cancer


The effectiveness of alkalization therapy was explored in a cohort of stage 4 pancreatic cancer patients. This therapy, which included an alkaline diet, sodium bicarbonate, and/or sodium potassium citric acid administration alongside standard chemotherapy, was retrospectively assessed through urine pH measurements as a marker of whole-body alkalization. In the study of 98 patients, those with a mean urine pH over 7.5 showed a median overall survival (OS) of 29.9 months, longer than the 15.2 months and 8.0 months observed in patients with mean urine pHs of 6.5 to 7.5 and under 6.5, respectively (*p* = 0.0639) (Figure 2) [83]. These findings suggest that alkalization therapy could influence prognosis in a urine pH-dependent manner, potentially extending survival in patients typically faced with poor prognoses.


(ii)Small cell lung cancer


A study investigated the combined effects of alkalization therapy and intravenous vitamin C on chemotherapy outcomes in patients with small-cell lung cancer (SCLC). The intervention group (n = 12), which received both treatments in addition to chemotherapy, was compared retrospectively with a control group (n = 15) that received chemotherapy alone. The intervention group exhibited a significantly higher mean urine pH (7.32 ± 0.45) compared to the control group (6.44 ± 0.74; *p* < 0.005), and their median OS was 44.2 months, markedly longer than the 17.7 months in the control group (*p* < 0.05) (Figure 3) [78]. This study indicates that alkalization therapy, when combined with vitamin C, may significantly improve outcomes in SCLC patients.


(iii)Hepatocellular carcinoma


This study focused on patients with hepatocellular carcinoma (HCC) who received alkalization therapy in combination with standard treatments. Among the 29 patients retrospectively analyzed, the group with a mean urine pH ≥ 7.0 (n = 12) demonstrated a median OS from the start of alkalization therapy that was not reached, significantly outperforming the 15.4 months median OS observed in patients with a mean urine pH < 7.0 (n = 17, *p* < 0.05) (Figure 4) [79]. These results suggest that alkalization therapy may lead to more favorable outcomes in HCC patients, particularly those achieving higher urine pH levels following the therapy.

As described above, alkalization therapy appears to offer significant benefits for pancreatic cancer, small cell lung cancer, and hepatocellular carcinoma, particularly when integrated with standard cancer treatments. By altering the acidic TME, this therapy potentially extends patient survival and enhances the effectiveness of conventional treatment modalities. However, it is important to note that these findings are based on retrospective studies. Regarding the side effects of alkalization therapy, no significant adverse effects were observed in these retrospective studies. Additionally, a clinical trial has shown that long-term sodium bicarbonate administration at 0.17 g/kg/day for 90 days was safe and effectively increased urine pH, indicating its safety and efficacy [84]. In summary, alkalization therapy shows promise for several cancers and is safe based on initial trials, but further randomized controlled trials are needed to confirm its efficacy.

## 7. Discussion

In our view, the fundamental cause of cancer lies in the mitochondria present within the cells of eukaryotic organisms [2,3,12,13,14]. In his seminal paper “On the Origin of Cancer Cells”, Warburg stated, “If we could understand how the damaged respiration and excessive fermentation in cancer cells occur, we could grasp their origin” [12]. As he suggested, many cancers survive predominantly on glycolysis. Remarkable experiments inducing cancer under oxygen-deficient conditions were reported by Goldblatt and Cameron [14]. When cardiac fibroblasts cultured in tissue were subjected to intermittent oxygen deprivation over an extended period, transplantable cancer cells eventually developed. In contrast, control cultures maintained without oxygen deprivation did not produce cancer cells. Warburg also posited, “If the failure of respiration causes cancer, then this failure must be irreversible” [12]. Vaupel and Multhoff stated, “In most tumors, the Warburg effect represents an essential part of ‘selfish’ metabolic reprogramming, with the earliest genetic abnormalities being the overexpression of hypoxia-inducible factor-1 (HIF-1)” [20].

In summary, localized hypoxia occurs when oxygen supply becomes intermittently obstructed. However, since nutrients in the bodily fluids (such as glucose) continue to be supplied, cells in such an environment are placed in a state where nutrient supply remains sufficient despite the irregular decrease in oxygen supply. It is believed that the fundamental cause of cancer originates from these conditions, where some signals are transmitted from the mitochondria to the nucleus, allowing the cell to survive through a capricious metabolism. These signals transmitted from the mitochondria to the nucleus are known as retrograde signal responses [17,18]. Cancer cells thus formed gradually carve out a niche within the TME. Due to the activation of glycolysis in cancer cells, which produces a large amount of acidic products, cancer cells expel protons outside to maintain an alkaline intracellular pH. The activity of the NHE-1 is enhanced in cancer cells. It has been reported that NHE-1 ceases its activity when the extracellular pH reaches 7.5 [85]. When the TME becomes acidic, cancer cells have been reported to exhibit resistance to various treatments [32,86,87,88].

Conversely, neutralizing the acidic TME to a neutral or slightly alkaline state halts the proliferation of cancer cells, making various treatments effective. We have defined such treatment strategies as “Alkalization therapy”, which we have already reported [48,49]. As our bodies are dissipative structures, ‘entropy’ accumulates under normal living conditions. The importance of diet in the treatment of diseases, especially cancer, has been emphasized. With this understanding, we have established “Alkalization therapy” as a strategy against cancer. The sole method to alkalize the body, as suggested by Remer and Mantz, is through the consumption of fruits and vegetables [89]. In some cases, alkalization therapy alone can halt the growth of cancer or even lead to its eradication [49]. The fundamental concept of this therapy targets the phenomenon of ‘entropy’ within the body. Based on this concept, we have established “alkalization therapy” for cancer treatments.

Limitations of this review include the following points. While in vitro and in vivo studies have shown that alkalization of the TME is associated with suppression of cancer metabolism, the effectiveness in clinical settings remains unclear [22,32,34,51,68,88,90]. The clinical efficacy of alkalization therapy has only been suggested by a few retrospective studies and case reports [48,49,78,79,82,83]. Therefore, it is necessary to conduct randomized controlled trials across various cancer types to investigate the effectiveness of alkalization therapy for cancer.

## 8. Conclusions

In conclusion, cancer development is likely driven by a condition of intermittent oxygen deprivation that pushes cells towards a glycolytic metabolism, leading to carcinogenesis. This process results in the acidification of the tumor microenvironment (TME), promoting cancer malignancy and drug resistance. Alkalization therapy, aimed at neutralizing this acidity, may offer a promising strategy for improving cancer treatment in some cancer types by potentially stopping or eliminating tumor growth by targeting the body’s entropy. However, further investigation is necessary to evaluate its efficacy and determine its applicability across various cancers.

## Figures and Tables

**Figure 1 biomedicines-12-02197-f001:**
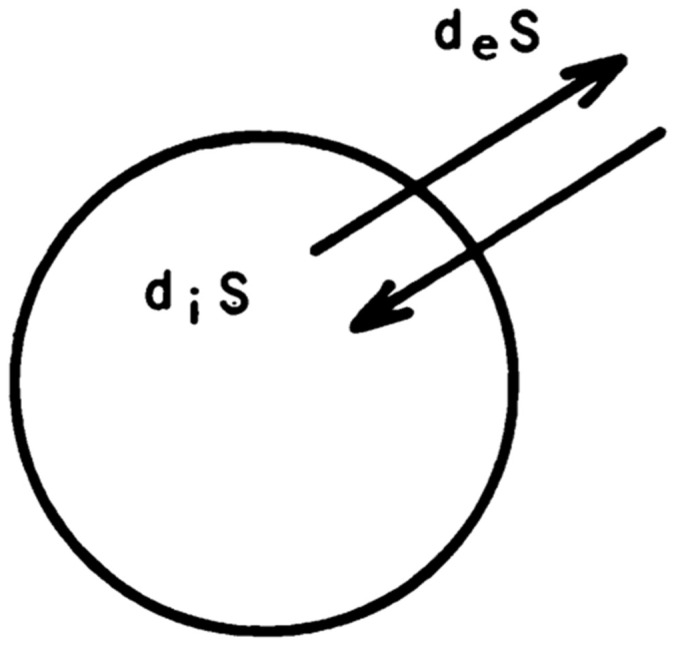
The exchange of entropy between the outside and the inside (Reproduced by permission of “Science”. For further details see Ref. [45]). The diagram shows the entropy change (dS), divided into entropy transferred across the system boundary (deS) and entropy generated within the system (diS). While diS is typically non-negative as per the Second Law of Thermodynamics, dissipative structures can exhibit reductions in dS, demonstrating negative entropy phenomena.

**Figure 2 biomedicines-12-02197-f002:**
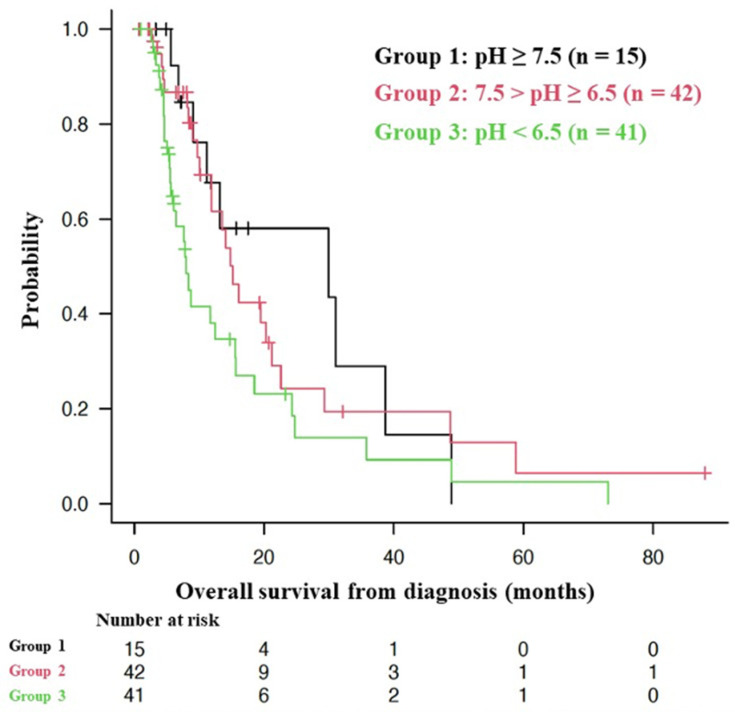
Association between overall survival and urine pH in stage 4 pancreatic cancer patients [Cited from [83]]. Kaplan–Meier curves show the OS for three groups based on initial urine pH: Group 1 (pH ≥ 7.5, n = 15) with a median OS of 29.9 months, Group 2 (7.5 > pH ≥ 6.5, n = 42) with a median OS of 15.2 months, and Group 3 (pH < 6.5, n = 41) with a median OS of 8.0 months.

**Figure 3 biomedicines-12-02197-f003:**
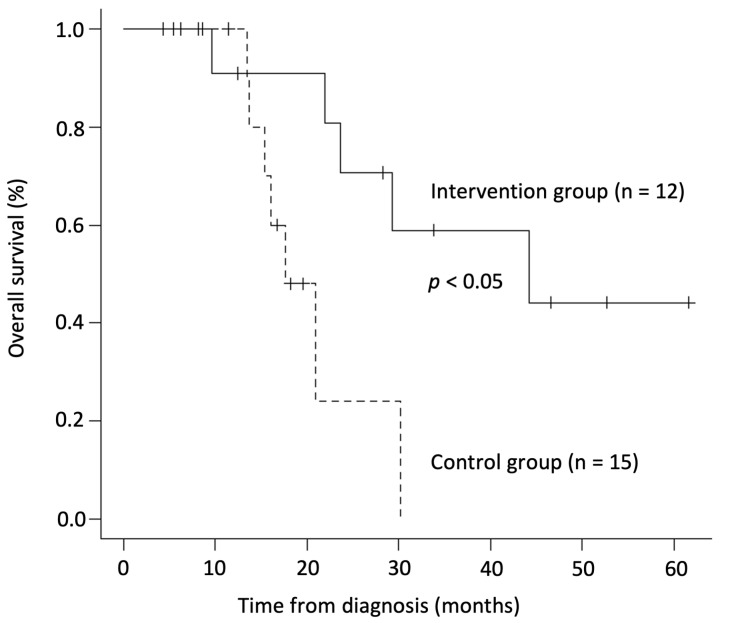
Overall survival comparison in small cell lung cancer patients [Cited from [78]]. Kaplan–Meier curves show the OS from diagnosis for small cell lung cancer patients. The intervention group, treated with chemotherapy, alkalization therapy, and vitamin C (n = 12), shows a median OS of 44.2 months, compared to 17.7 months for the control group, which received chemotherapy only (n = 15).

**Figure 4 biomedicines-12-02197-f004:**
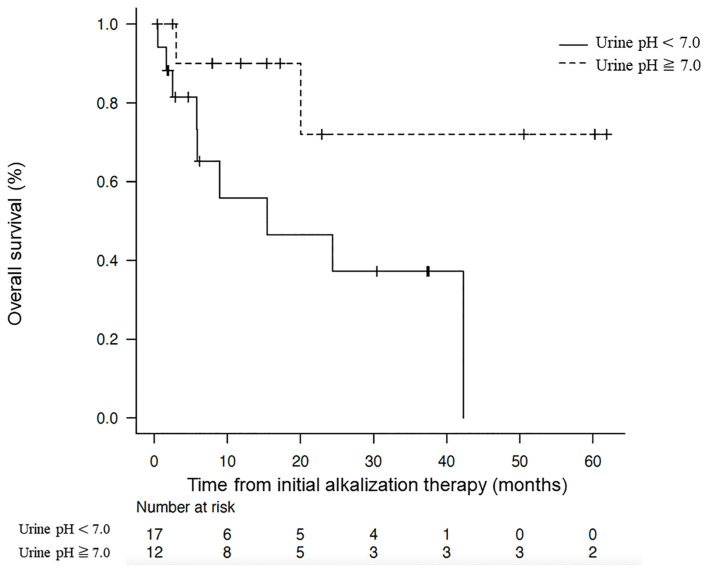
Association between overall survival and urine pH in hepatocellular carcinoma patients [Cited from [79]]. Kaplan–Meier curves compare the OS from the start of alkalization therapy in hepatocellular carcinoma patients categorized by mean urine pH levels. Patients with a mean urine pH ≥ 7.0 (n = 12) showed a median OS that was not reached, significantly longer than the 15.4 months median OS for those with a mean urine pH < 7.0 (n = 17).

## Data Availability

Not applicable.

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
