# Peer review of "How Does Cancer Occur? How Should It Be Treated? Treatment from the Perspective of Alkalization Therapy Based on Science-Based Medicine"

_biomedicines, 2024, doi:10.3390/biomedicines12102197_

Round 1
Reviewer 1 Report
Comments and Suggestions for Authors
The main focus of this article is to examine the correlation between mitochondrial dysfunction and the progression of cancer, with a particular emphasis on metabolic shifts that facilitate tumor growth. The article appears to be intriguing and acceptable, however, the author should consider addressing the following concerns.
1. In the introduction section, it is recommended for the author to incorporate a paragraph elucidating the primary content of the article, delineating its focal point, highlighting its novelty in comparison to previous reviews, and specifying any research pertinent to the title that has been excluded from this article.
2. The relationship between the tumor microenvironment and tumor immunity is closely intertwined, and the author can provide concise descriptions on this interconnectedness. The literature listed below provides a comprehensive description of the tumor microenvironment.
[1] A. Gu, J. Li, S. Qiu, S. Hao, Z.-Y. Yue, S. Zhai, M.-Y. Li, Y. Liu, Pancreatic cancer environment: From patient-derived models to single-cell omics. Mol. Omics 2024, 20, 220–233. DOI: 10.1039/D3MO00250K
[2] Guo J, Wang MF, Zhu Y, Watari F, Xu YH, Chen X. Exploitation of platelets for antitumor drug delivery and modulation of the tumor immune microenvironment. Acta Materia Medica. 2023, 2(2): 172-190. DOI: 10.15212/AMM-2023-0005
[3] Huo RW, Zhan MJ, Zhu B, Zhi Q. Annual advances of traditional Chinese medicine on tumor immunity regulation in 2021. Tradit Med Res. 2022;7(6):56. doi:10.53388/TMR20220615001
3. The authors require a more comprehensive introduction to elucidate the correlation between entropy and tumor therapy, as well as incorporate additional scholarly references.
4. The authors discuss the clinical application of science-based medicine in pancreatic cancer, non-small cell lung cancer, and hepatocellular carcinoma; however, it remains unclear whether there are other instances of tumor utilization.
5. The therapeutic efficacy of this treatment on gastrointestinal tumors is comparable to that observed in other tumor types.
6. Is there any ongoing research pertaining to leukemia?
7. As a review, this manuscript has too few references and needs to add a large number of references
Reviewer 2 Report
Comments and Suggestions for Authors
The present manuscript by Hamaguchi et al., titled "How Does Cancer Occur? How Should It Be Treated? Treatment from the Perspective of Alkalization Therapy Based on Science-Based Medicine," aims to investigate the relationship between mitochondrial dysfunction and cancer progression, highlighting the metabolic alterations that promote tumor growth. The authors explore how metabolic reprogramming in cancer cells, especially the Warburg effect, fosters an acidic microenvironment that facilitates cell proliferation, invasion, and resistance to conventional treatments.
Below are points the authors need to address:
- The manuscript lacks robust randomized clinical trials to support the conclusions on the efficacy of alkalization therapy. The evidence is mainly retrospective, which limits the generalizability of the results.
- I find the use of urinary pH as an indicator of systemic alkalization questionable, as it may not accurately reflect the tumor microenvironment. It is suggested to better justify this choice or explore other, more direct markers.
- Although the article discusses tumor aggressiveness reduction through alkalization, details are missing about the molecular mechanisms explaining this relationship. This aspect needs to be included in the introduction and discussion of the manuscript.
- There is no adequate discussion of the potential adverse effects of alkalization therapy, such as the risk of metabolic alkalosis. It is important to include this analysis.
- The article generalizes preliminary results of alkalization therapy across different cancer types without robust evidence to support this generalization. It should be verified if this can be described in relation to certain tumor types that already show evidence in the literature.
- The introduction contains broad information on basic biology and mitochondrial evolution that diverts focus from the central theme. It would be good to shorten it and focus more on cancer metabolism.
- The authors should suggest more controlled prospective studies to validate the preliminary findings on the efficacy of alkalization in cancer treatment.
- The relationship between alkalization and improved survival should be discussed with more caution, highlighting the limitations of the evidence and the need for additional studies to confirm the findings.
- The conclusion should be adjusted to avoid exaggerated promises about the benefits of alkalization, recognizing the current limitations of the evidence.
Minor revisions to the English language are needed
Round 2
Reviewer 2 Report
Comments and Suggestions for Authors
All the suggested changes have been successfully implemented, and after a thorough review, I recommend that the manuscript is ready for publication. The work now presents a clearer structure, with improvements that enhance the scientific quality and the contribution of the study to the field.
Comments on the Quality of English LanguageThe quality of the English language is adequate. The text is clear, well-structured, and effectively communicates the key points of the study. Minor grammatical or stylistic improvements could be made, but overall, the language is appropriate for publication